# Mutual Regulation between Redox and Hypoxia-Inducible Factors in Cardiovascular and Renal Complications of Diabetes

**DOI:** 10.3390/antiox11112183

**Published:** 2022-11-04

**Authors:** Carla Iacobini, Martina Vitale, Jonida Haxhi, Carlo Pesce, Giuseppe Pugliese, Stefano Menini

**Affiliations:** 1Department of Clinical and Molecular Medicine, “La Sapienza” University, 00189 Rome, Italy; 2Department of Neurosciences, Rehabilitation, Ophthalmology, Genetic and Maternal Infantile Sciences (DINOGMI), Department of Excellence of MIUR, University of Genoa Medical School, 16132 Genoa, Italy

**Keywords:** advanced glycation end products, atherosclerosis, diabetic kidney disease, inflammation, methylglyoxal, prolyl hydroxylase domain proteins, reactive oxygen species, sirtuin-1, sodium glucose co-transporter 2 inhibitors, Warburg effect

## Abstract

Oxidative stress and hypoxia-inducible factors (HIFs) have been implicated in the pathogenesis of diabetic cardiovascular and renal diseases. Reactive oxygen species (ROS) mediate physiological and pathophysiological processes, being involved in the modulation of cell signaling, differentiation, and survival, but also in cyto- and genotoxic damage. As master regulators of glycolytic metabolism and oxygen homeostasis, HIFs have been largely studied for their role in cell survival in hypoxic conditions. However, in addition to hypoxia, other stimuli can regulate HIFs stability and transcriptional activity, even in normoxic conditions. Among these, a regulatory role of ROS and their byproducts on HIFs, particularly the HIF-1α isoform, has received growing attention in recent years. On the other hand, HIF-1α and HIF-2α exert mutually antagonistic effects on oxidative damage. In diabetes, redox-mediated HIF-1α deregulation contributes to the onset and progression of cardiovascular and renal complications, and recent findings suggest that deranged HIF signaling induced by hyperglycemia and other cellular stressors associated with metabolic disorders may cause mitochondrial dysfunction, oxidative stress, and inflammation. Understanding the mechanisms of mutual regulation between HIFs and redox factors and the specific contribution of the two main isoforms of HIF-α is fundamental to identify new therapeutic targets for vascular complications of diabetes.

## 1. Introduction

Diabetes represents a growing global health issue that significantly contributes to premature mortality, morbidity, and disability, and poses a tremendous economic burden to national health systems [1,2]. Most of the harms and costs related to diabetes are due to its long-term microvascular (nephropathy, retinopathy, and neuropathy) and macrovascular (coronary, cerebrovascular, and peripheral artery disease) complications [3]. Redox imbalance, dysregulated hypoxia-inducible factor (HIF) signaling, and mitochondrial abnormalities have all been implicated in vascular tissue damage associated with diabetes.

The role of reactive oxygen species (ROS) in diabetic complications has been recently reviewed [4,5]. Hyperglycemia and other diabetes-related metabolic abnormalities cause ROS overproduction in endothelial cells and perivascular cells of large and small vessels [6]. Although oxidative metabolism in mitochondria has long been considered the main source of superoxide overproduction in the setting of hyperglycemia [7], other cytosolic and plasma membrane oxidoreductases, such as NAD(P)H oxidases (NOXs), cycloxygenase-2, myeloperoxidase, etc., provide a substantial contribution to ROS production in diabetic tissues [4]. In addition to hyperglycemia, several other diabetes-associated stimuli may promote ROS formation and redox abnormalities, including accumulation of advanced glycation end products (AGEs) and their carbonyl precursors, inflammation, dyslipidemia, and upregulation of the renin-angiotensin system (RAS) [4,5]. Hence, oxidative stress is considered a key contributor to the pathogenesis of diabetes-related end-organ damage [8,9,10]. According to these premises, antioxidants might be predicted to protect against diabetic vascular complications. In fact, antioxidant therapies have failed to provide substantial clinical benefit [11]. This paradox might be explained by recent findings that, besides representing an important pathomechanism, oxidative stress and ROS are also part of essential signaling networks modulating different cellular functions, such as energetic metabolism, growth, survival, angiogenesis, vascular tone regulation, etc. [4]. Consistent with a regulatory function in fundamental cellular processes, physiological enzymatic sources of ROS such as NOXs are evolutionarily conserved enzymes whose only identified function is to produce ROS [11]. In this perspective, ROS are not merely toxic and metabolic waste products that need to be removed to prevent cellular damage. The final result of ROS action does not depend exclusively on their absolute concentration (i.e., the so-called “redox balance hypothesis”), but also on the amount and type of ROS produced in specific subcellular compartments and, above all, on their interactions with pathways and systems that are closely related to energy and oxygen metabolism.

HIFs are master regulators of oxygen homeostasis, playing a key role in the adaptive regulation of energy metabolism in mammalian tissues [12]. The HIF pathway has been implicated in the development of cardiovascular and renal complications of diabetes [13] (Figure 1).

However, the net effect on HIF signaling of the numerous metabolic stressors associated with diabetes, which include hyperglycemia, carbonyl and oxidative stress, and hypoxia, has not been conclusively defined, and several disputes exist among researchers on this issue [13,14,15]. In fact, HIFs have been shown to have both beneficial and detrimental roles in diabetic kidney disease (DKD) [16,17,18,19], retinopathy [20,21,22], and cardiovascular disease [23,24]. Possible reasons for these contrasting results include the heterogeneity of the experimental models and the methods used for modulating the expression, stability, and transcriptional activity of HIFs: for example, the type and combination of metabolic stressors and the presence or absence of hypoxia. Furthermore, there is evidence that the two main members of the HIF family (i.e., HIF-1α and HIF-2α) have different effects on redox balance, inflammation, and fibrosis [24], which are the major pathogenic mechanisms of tissue damage associated with diabetes. Ultimately, these discrepancies are likely the result of the complex intersection of HIF biology with metabolic disorders.

Among the diabetes-related factors potentially influencing HIF homeostasis, changes in the redox status and signaling have definitely been shown to affect HIF protein stability and activity [25,26]. Considering also that HIFs play a key role in the regulation of cellular glucose metabolism and redox homeostasis [27], and that the activity of both HIF proteins and redox factors is strictly connected to the availability of oxygen, glucose, and their metabolisms, a mutual interaction between these two integrated cellular energetic systems in diabetes appears inevitable.

## 2. The HIFs Family

HIFs are heterodimeric transcription factors acting as master regulators of oxygen homeostasis. By regulating a large battery of genes and activating a broad range of transcriptional responses, HIFs match oxygen supply and demand in tissues [12] (Figure 2).

The HIF family includes three hypoxia-regulated α-subunits (HIF-1α, HIF-2α, and HIF-3α), which are associated with a constitutively expressed β-subunit (HIF-1β). HIF-1α and HIF-2α are the principal HIF-α isoforms. Despite an identical consensus recognition sequence in DNA, HIF-1α and HIF-2α do not compete for binding sites and activate common as well as unique target genes [28,29]. Therefore, HIF-1α and HIF-2α may function more independently than previously thought, suggesting the need for isoform-specific HIF inhibitors for specific therapeutic indications [30]. Little is known about HIF-3α. This isoform has been considered a negative regulator of the hypoxic response by competing with the other two isoforms [31], but its role has yet to be fully defined.

In the presence of oxygen, iron, and 2-oxo-glutarate (α-ketoglutarate), the HIF-α subunits are continuously targeted for proteasomal degradation by three HIF prolyl hydroxylase domain proteins (PHD1-3). PHD2 plays a prominent role in the regulation of the HIF signaling pathway [32,33] as it is the rate-limiting enzyme for the degradation of the HIF-1α subunit in normoxia [32]. PHDs are α-ketoglutarate—dependent dioxygenases that rapidly hydroxylate HIF-α subunits at two proline residues (Pro-402 and Pro-564 in HIF-1α) in the presence of oxygen. Prolyl hydroxylated HIF-α subunits are recognized by the von Hippel–Lindau (VHL), a ubiquitin E3 ligase complex that marks HIF-α proteins for proteasomal degradation [33]. In addition to the negative regulation of protein stability by PHDs, the transactivation function of HIF-1α and HIF-2α is inhibited by factor-inhibiting HIF-1 (FIH-1) [34], another oxygen sensor that hydroxylates HIF-α on an asparagine residue (N-803) in its C-terminal transactivation domain [35]. This oxygen-dependent post-translational modification prevents the interaction of HIF-α proteins with transcriptional co-activators, such as cAMP response element-binding protein (CBP) and E1A binding protein p300 [36,37]. Therefore, PHD and FIH enzymes ensure the full repression of the HIF pathway in normoxia by controlling the degradation and transcriptional activity, respectively. Conversely, insufficient levels of oxygen (i.e., hypoxia) inhibit hydroxylation, impair HIF-α proteasome degradation, and favor HIF-α stability and dimerization with the HIF-1β subunit, which facilitates translocation to the nucleus, the recruitment of transcriptional coregulators, and binding to a hypoxia response element (HRE) in various target genes. Overall, oxygen-dependent hydroxylations on proline and asparagine residues modulate HIF-α stability and activity so that changes in oxygen availability are transduced to the nucleus as changes in HIF-dependent transcriptional activity.

In addition to hypoxia, HIF-α proteins respond to oxygen-independent stimuli, including growth factors, cytokines, vasoactive peptides, coagulation factors, and hormones such as insulin [38,39,40,41,42]. Many metabolic factors, including glucose and their glycolytic metabolites, can also affect the HIF pathway in opposite directions [43,44], possibly depending on the availability of oxygen. The functions of HIFs in oxygenated cells and the mechanisms involved in the regulation of HIF signaling by metabolic stimuli in normoxic or hypoxic conditions are not yet fully understood. Nevertheless, the modulation of HIF activity by metabolic changes suggests its contribution in the reactive adaptation to dynamic microenvironment variations, and also in the absence of alterations in oxygen tension. Regarding the mechanism(s), most of the nonhypoxic stimuli capable of modifying HIF activity induce ROS production as part of their signaling cascade [45]. Therefore, a role of ROS in HIF-α stabilization and activity has been hypothesized, particularly under normoxia conditions.

## 3. Redox Regulation of HIFs: Role of Diabetes-Related Stimuli

Hyperglycemia-driven mitochondrial dysfunction [46], the over activation of redox-enzymes in vascular and inflammatory cells [4], and glucose autoxidation catalyzed by trace amounts of transition metal ions [47], such as iron and copper, are deemed the major source of ROS in diabetic vascular tissues. In addition, as part of the cardiometabolic syndrome, the activation of the RAS system triggers ROS formation via angiotensin (Ang) II type 1 receptor stimulation, thus contributing to endothelial dysfunction and diabetes-associated cardiorenal diseases [48]. Compelling evidence exists that ROS take part in the regulation of HIF-α [25,49,50,51,52] (Figure 3).

While the role of mitochondrial ROS under hypoxia has long been debated with no clear consensus, there is considerable evidence in support of the normoxic stabilization of HIF-α proteins by ROS. The first demonstration that oxidative stress contributes to HIF-α stabilization under normoxia comes from studies showing that the addition of exogenous ROS, or ROS-generating enzymes, is sufficient to stabilize HIF-1α protein and induce the transcription of HIF target genes, such as vascular endothelial growth factor (VEGF) [49,51,53]. Moreover, all these effects were attenuated by cellular treatments with antioxidant compounds such as catalase, glutathione, N-acetyl cysteine, and vitamins E and C [54,55,56,57,58,59,60,61,62,63,64]. HIF-2α stability is similarly affected by the addition of ROS [53], a finding confirmed and extended by other studies showing that NOX4-derived hydrogen peroxide (H_2_O_2_) increases HIF-2α stability and its transcriptional activity [26,64]. Approaches directed at increasing the levels of the superoxide anion by silencing or the pharmacological inhibition of superoxide dismutase (SOD) also demonstrated a role for ROS in HIF-1α expression and protein accumulation in normoxia [65,66], but not in hypoxia [65]. Overall, these findings implicate ROS in normoxic HIF-α stabilization and activity by showing that HIF-α stability is sensitive to the redox status in many cell types and that ROS such as superoxide and H_2_O_2_ are sufficient for activating the HIF pathway.

As far as the mechanism, ROS were shown to inhibit PHD activity and VHL binding to HIF-1α under normoxia [65,67]. It has been suggested that ROS may oxidize the iron co-factor of the PHD enzymes, thus inhibiting their activity [68]. Hence, sustained ROS production may reduce the cellular pool of ferrous ion, which is required for the full activity of PHD enzymes [69,70]. Consistently, adding cobalt ions to cells with normal oxygen tension mimics a hypoxia response through a competition mechanism between cobalt and ferrous ions for binding with the enzyme active site, thus impairing PHDs’ activity and HIF-α hydroxylation [71]. Likewise, the iron chelator desferrioxamine causes HIF-α proteins to accumulate in a dose-dependent manner [72,73,74].

In addition to affecting HIF-α protein levels by the disruption of PHD and/or FIH-1 activity, ROS have also been suggested to mediate the transcriptional and translational regulation of HIF-α via indirect mechanisms, including phosphoinositide 3-kinase (PI3K)-Ak strain transforming (Akt) and extracellular signal-regulated kinase (ERK) pathways. A key role of ERK and PI3K/Akt signaling in ROS-mediated HIF-1α stabilization has been suggested in many disease conditions including neoplastic, ischemic, and inflammatory disorders [25,75]. Consistently, tyrosine kinase inhibitors were demonstrated to block HIF-α synthesis and activity in hypoxia, showing that protein phosphorylation plays an important role in HIF signaling [25,75]. In addition, many nonhypoxic stimuli, including insulin, thrombin, and growth factors were shown to stimulate the HIF response through ERK and PI3K/Akt signaling pathways in a redox-sensitive manner [39,54,56,76,77,78,79,80,81,82,83].

In diabetes, excessive ROS generation triggers the production of proinflammatory compounds, such as oxidized LDLs (oxLDLs) and advanced glycoxidation/lipoxidation end products (AGEs/ALEs), which may serve as additional nonhypoxic stimuli for HIF activation. The levels of these byproducts of glucose and lipid peroxidation are increased in the bloodstream and tissues of diabetic subjects, where they propagate metabolic and redox signals through interactions with several specific receptors, in particular the receptor for AGEs (RAGE) [84,85,86]. RAGE signaling elicits the activation of multiple intracellular signaling pathways, including the protein kinases Akt, ERK1/2 mitogen activated kinases, and c-Jun N-terminal kinase. Eventually, these signaling pathways result in the activation of redox-sensitive and proinflammatory transcription factors such as nuclear factor kappa B (NF-κB) and activator protein 1 (AP-1) [84,85,86,87]. Importantly, oxLDLs and AGEs/ALEs have been involved in the enhancement of HIF-α protein expression [88,89], stability [58,89,90], and transcriptional activity [91,92,93] in diabetic vasculopathy. Both hyperglycemia and AGEs/ALEs promote the protein accumulation of HIF-α proteins—mainly HIF-1α—and HIF activity in glomerular mesangial and renal tubular epithelial cells in in vitro and in vivo models of DKD [17,94]. Moreover, the AGE-RAGE signaling pathway augments ROS generation by NOX enzymes [95], thereby generating a vicious cycle that boosts oxidative stress, AGE/ALE production, and HIF signaling dysregulation. Altogether, these findings support the hypothesis of a role for the redox regulation of HIF activity by diabetes-related, nonhypoxic stimuli in the pathogenesis of cardiovascular and renal complications of diabetes.

Finally, a potential role for some microRNAs (miR) in the redox- and PI3K/Akt-dependent stabilization of HIF-1α has been proposed [25]. In particular, miR21 is induced by ROS [96] and plays a role in both PI3K-Akt and HIF-1α signaling activation [97]. Interestingly, miR21 has also been implicated in diabetes development and ROS-mediated damage [98] as well as DKD [99,100]. Together with the observation that PI3K and Akt signaling is altered by hyperglycemia [101], these findings suggest the involvement of a miR21-PI3K-Akt axis in the redox regulation of HIF in diabetic complications.

## 4. Modulation of Redox Homeostasis by HIFs in Response to Changes in Oxygen or Energy Substrate Availability

At physiological concentrations, ROS serve as critical signaling molecules supporting normal processes such as cellular proliferation, migration, and metabolic adaptation [102]. To give some examples, ROS signaling plays a critical role in the cells involved in innate and adaptive immunity [103,104] by ensuring a healthy immune system. Decreased ROS levels inhibit the activation of the immune system, leading to immunosuppression, whereas excessive ROS levels or prolonged ROS signaling can result in the hyperresponsiveness of the immune system through the release of inflammatory cytokines, leading to chronic inflammation and/or autoimmunity [105]. Physiological ROS levels are also required for stem cell maintenance and function. In fact, redox signaling plays an important role in tissue regeneration and repair [106,107]. It follows that redox status might be a potential therapeutic target in many oxidative stress-related diseases, including diabetic cardiovascular and renal diseases. However, targeting oxidative stress should consider the complexity of the ROS signaling network and its context-dependent specificity.

HIF signaling determines the balance between oxidative phosphorylation and glycolysis by regulating the amount of oxygen consumed by mitochondria in response to changing oxygen or substrate levels [108]. Reduced oxygen availability and/or an increase in substrate supply has profound effects on cellular energy metabolism and redox equilibrium [109,110]. Under conditions of hypoxia, mitochondrial electron transport becomes less efficient, leading to the increased generation of superoxide anions. To maintain redox homeostasis under hypoxic conditions, HIFs factors activate the transcription of target genes encoding proteins that serve to switch cells from oxidative to glycolytic metabolism and maintain redox homeostasis. More specifically, HIF signaling decreases mitochondrial superoxide generation by reducing glucose oxidation in the tricarboxylic acid cycle and enhances cellular antioxidant defenses, mainly by promoting the biosynthesis of glutathione, a major cellular antioxidant [27,111]. Excess glucose entry in insulin-independent cells such as endothelial, neuronal, and renal cells is believed to increase mitochondrial ROS generation through a mechanism similar to that of hypoxia [46], and recent evidence suggests that the repression of HIF-1α contributes to increased ROS production in diabetes [112]. All together, these effects may predict a pathological role of the mitochondrial superoxide anion and a protective role of HIF-1α accumulation in the pathogenesis of diabetic vascular complications. However, much of the current evidence does not support this conclusion and suggests that the role of oxidative stress and HIFs in the regulation of redox balance in cardiovascular and renal disease is much more complex than that.

Growing evidence from a small but increasing number of independent laboratories shows that, in normoxia, hyperglycemia is associated with a reduction, not an increase, in mitochondrial superoxide generation and that restoring mitochondrial superoxide levels protects against the progression of experimental DKD [43,113,114,115]. In addition, HIF-1α is upregulated and transcriptionally active in vascular, kidney, and inflammatory cells during glucose exposure in normoxic cultures [17,18,43,116,117], as well as in diabetic vascular and renal tissues [17,88,117,118]. Consistent with a negative role of HIF-1α, its downregulation protects from cell dysfunction, proinflammatory cell activation, and tissue damage induced by high glucose and toxic glucose metabolites [17,18,43,94,116,117]. On the one hand, these findings exclude a role for oxidative stress as the primary instigator of diabetic complications and suggest that ROS overproduction may be a downstream event resulting from earlier molecular changes induced by hyperglycemia or other metabolic stressors. On the other hand, they suggest a complex regulation of HIF signaling by metabolic and local tissue factors, including hyperglycemia [17,18,43,88,116], AGEs/ALEs accumulation [17,94], mechanical stress [119,120,121], elevated Ang II levels [122], and macrophage tissue infiltration [123]. By affecting HIF signaling, each of these metabolic and diabetes-associated factors, alone or in combination, may contribute to a molecular network that acts as the primary instigator of cardiovascular and kidney complications. In line with this assumption, a different hypothesis of diabetic complications, alternative to that of mitochondrial superoxide overproduction, proposes a link between glycolysis-derived carbonyl stress and the establishment of a Warburg effect. Both processes (i.e., carbonyl stress and increased aerobic glycolysis) are closely related to the intracellular metabolism of excess glucose and mitochondrial dysfunction, and are deemed to play a key role in the very early stages of diabetic complications [5,43,124,125,126] (Figure 4).

In more details, the carbonyl compound and AGE precursor methylglyoxal, which is a byproduct of glycolysis [127] and a major mediator of glucose toxicity to cardiovascular and renal tissues [128,129], stabilizes HIF-1α and induces a shift in energy production from mitochondrial respiration towards glycolysis under aerobic conditions (i.e., the Warburg effect), mainly by modifying and inhibiting PHD2 [43]. However, in contrast with iron-dependent PHD regulation, mitochondrial ROS are not involved in the methylglyoxal-mediated inhibition of PHD2 and consequent HIF-1α activation. Eventually, sustained HIF-1α activity, persistent glycolysis upregulation, and the shunting of glycolytic intermediates into glycolysis side branches feed the main pathways of hyperglycemic damage, leading to inflammation and oxidative stress [5,43,46]. In this hypothesis, the metabolic reprogramming driven by HIF-1α in response to the increased glycolytic flux and accumulation of toxic glucose metabolites [43,125,126,130] is the early event, upstream of the proinflammatory and oxidative response of immune and endothelial cells [43,131,132]. Secondary to these alterations and to the initial vascular injury, local tissue hypoxia and an impaired response of HIFs to oxygen level changes may also participate in the progression of diabetic complications [14,16,44,133].

In a more general context of persistent HIF-α induction, this mechanistic model may explain the dual effect of PHD inhibition in the development of chronic kidney disease (CKD), depending on the timing of the administration: deleterious by an early treatment, when HIF-α signaling contributes to the pathological response to a metabolic stressor; beneficial by a late treatment, when the response to hypoxia is impaired [134]. Interestingly, the early deleterious treatment with PHD inhibitors was associated with HIF-1α activation and increased proinflammatory and profibrotic signaling, whereas the late beneficial treatment preferentially activated HIF-2α and induced VEGF and erythropoietin (EPO) expression, suggesting that the effects of PHD inhibition also depend on the activated isoform of HIF-α [134].

In fact, to complicate matters further, HIF-α isoforms have both overlapping and unique target genes [28,29]. Although the effects of HIF-1α and HIF-2α on oxygen consumption and glucose metabolism are concordant, these isoforms exert mutually antagonistic effects on other aspects of cellular homeostasis, including redox balance and the modulation of proinflammatory and profibrotic pathways [24] (Figure 5).

In particular, the interplay of HIF-1α and HIF 2α helps establish the set point for the redox state of cells by regulating both the production and disposal of ROS. For example, increased levels of HIF-1α in HIF-2α hypomorphic mice is accompanied by a higher expression of the cytosolic enzyme NOX2 and a shift toward the oxidation of the intracellular redox state. Conversely, the increased expression of HIF-2α in HIF-1α hypomorphic mice is accompanied by an increased expression of the antioxidant enzyme SOD and a reduced intracellular redox state [135,136]. The balance between HIF-1α and HIF-2α may also establish the set point for inflammation and fibrosis in various organs, including the heart, vasculature, and kidney [134,137,138]. HIF-1α promotes M1 macrophage polarization, the induction of proinflammatory cytokines, and T cell activation [131,132], whereas HIF-2α has different and even opposite effects on these inflammatory responses [138,139,140,141,142]. Similarly, HIF-1α transactivates genes that encode for profibrotic chemokines and collagen deposition [143], whereas HIF-2α promotes extracellular matrix degradation [144]. According to these findings, the imbalance between HIF-1α and HIF-2α has been suggested to contribute to the onset and progression of cardiovascular and renal diseases [24].

## 5. Dysregulated HIF Signaling and Redox Homeostasis in Cardiovascular and Renal Complications of Diabetes: Insights from Pharmacological HIF Modifiers and Perspectives for Future Research

A prompt and coordinated response to ischemia or hypoxia by HIF-1α is necessary for promoting angiogenesis, tissue repair, and the regeneration of cardiovascular and renal tissues [145,146,147,148]. Similar beneficial effects in ischemic coronary and renal arteries have also been reported for HIF-2α activation [149,150]. However, the sustained upregulation of HIF-1α exerts negative effects via the activation of redox-dependent and inflammatory signaling pathways, irrespective of the hypoxic or normoxic conditions [58,119,120,151,152,153,154,155,156].

As anticipated above, many nonhypoxic stimuli associated with diabetes and related metabolic disorders may potentially alter the balance between HIF-1α and HIF-2α, both acting directly or through the activation of the nutrient/energy sensors that are involved in the dynamic adaptation to energetic and redox variations [15,24]. Therefore, inflammation and redox imbalance in diabetic vascular tissues may be a consequence of deranged HIF signaling by metabolic and local tissue factors.

Cardiovascular and renal complications of diabetes involve glyco- and lipoxidation reactions, arterial and glomerular hemodynamic modifications, inflammation, and hypoxia in target tissues. Together with hyperglycemia [17,18,43,116], AGEs/ALEs accumulation [58,88,89,90,91,92,93], tissue infiltration by inflammatory cells [123], increased Ang II levels [157], and blood-flow-induced mechanical forces [120,143] promote the activation of HIF-1α and consequent energetic metabolic reprogramming. HIF-1α inhibition ameliorates experimental DKD by attenuating glomerular hypertrophy, glomerular and interstitial fibrosis, and urinary albumin excretion, at least in part through the reduction in NOX4-dependent ROS production [19]. Similarly, HIF-1α deficiency in myeloid cells or endothelial cells reduces lesion burden in mouse models of atherosclerosis by restraining inflammation, oxidative stress, and glycolytic rate [123,158]. Furthermore, clinical findings suggest a role of HIF-1α in vascular calcification in diabetic patients [159]. Conversely, the activation of adipose HIF-2α protects, whereas HIF-2α deficiency exacerbates atherosclerosis [160]. Unfortunately, the role of HIF-2α in vascular complications of diabetes has been much less investigated than that of HIF-1α and, to our knowledge, only a couple of studies have evaluated the role of both HIF-α isoforms in parallel [161,162]. This is a research gap that needs to be filled, as evidence exists that the mutual antagonism between HIF-1α and HIF-2α controls the set points for redox balance, inflammation, and fibrosis [134,135,136,137,138]. Considering that the balance between HIF-1α and HIF-2α, and its changes over time, have been suggested to affect the development of cardiovascular and renal disease [24,134], this is a promising area for future research into diabetic complications.

### 5.1. PHD Inhibitors

HIF-α protein stabilization by PHD inhibition with roxadustat (FG-4592), a novel oral PHD inhibitor, has been claimed to increase the production of EPO and treat the anemia of CKD in clinical trials [163]. However, the first Cochrane review on this subject concluded that HIF stabilization has uncertain, if any, benefits on cardiovascular and renal outcomes in CKD patients, regardless of the presence of diabetes [164]. Experimental findings suggest that HIF-α induction by roxadustat, or the classic hypoxia mimetic cobalt chloride, can counteract diabetes-associated alterations in renal energy metabolism [165], protect against metabolic disorders and DKD [165,166,167,168], and prevent the progression of atherosclerosis [162]. Nonetheless, from these studies, it is unclear whether the protection afforded by PHD inhibition is mechanistically linked to the amelioration of metabolic abnormalities, activation of the HIF pathway, or both. What is more, these studies, except one [162], did not investigate the role of HIF-2α, which is the main isoform responsible for the HIF-mediated regulation of metabolism [162,169] and for the erythropoiesis-promoting effect of PHD inhibitors [170,171]. The findings that these drugs affect the activity of both HIF-1α and HIF-2α, and that HIF-1α induction by diabetes or PHD2 ablation has been shown to promote fibrogenesis and renal vascular remodeling [19,156,172] as well as sensitize kidney tissues and cells to hypoxia-induced fibrosis [173], suggest that the protection provided by these hypoxic mimetics most likely involves HIF-2α augmentation [24,171,174,175].

PHDs differ with respect to their effects on HIF-1α or HIF-2α. The inhibition of PHD2 acts primarily to increase the activity of HIF-1α [176], whereas PHD3 works preferentially on HIF-2α [24,177,178]. In general, hypoxia mimetics that increase EPO synthesis and treat anemia (e.g., cobalt chloride and roxadustat) mainly act by inhibiting PHD3 [30,176,179]. However, the degree of selectivity has not been assessed in most of the inhibition studies. Therefore, the possibility that these inhibitors suppress more than one isoform of PHD depending on dose, local tissue factors, or the stage of the disease cannot be ruled out. The failure to measure the effect of these drugs on HIF1-α and HIF2-α isoforms in parallel further complicates the understanding of their mechanism(s) of action in protecting against experimental organ injury.

Concerning the possibility of using PHD inhibitors in the treatment of the cardiovascular and renal complications of diabetes, it is important to first evaluate and carefully weigh the pros and cons, particularly of using these drugs in prevention and early intervention strategies. In fact, as mentioned above, PHD inhibition has dual effects on both HIF signaling and CKD depending on the timing of the intervention [134]. In particular, the early treatment with PHD inhibitors preferentially induces the upregulation of HIF-1α, whereas HIF-2α induction is better achieved by a late treatment, when the disease process is already established [24,134,150]. In ischemic kidney injury, the activation of HIF-2α promotes the resolution of inflammation and favors oxidative damage protection [150]. Conversely, the sustained activation of HIF-1α promotes oxidative-stress-mediated renal fibrosis, particularly in the diabetic kidney [19,156]. Therefore, before clinically testing the therapeutic effects of PHD inhibitors on cardiovascular and renal complications of diabetes, further experimental investigations are required to identify the optimal administration time window to avoid negative or unwanted effects.

### 5.2. Sirtuin-1 Activators

Sirtuin-1 (SIRT1) is a nicotinamide adenine dinucleotide (NAD+)-dependent enzyme that deacetylates transcription factors, thus promoting the expression of genes encoding proteins that function in the cellular stress response. SIRT1 regulates metabolism, stress resistance, and longevity by serving as both an energy- and redox-sensing molecule [180]. SIRT1 is also an important modulator of HIF-α isoforms, acting as a HIF switch that favors the activity of HIF-2α over that of HIF-1α [181]. Therefore, changes in SIRT1 activity may link metabolic and redox signaling alterations to an HIF-1α/HIF-2α imbalance. Obesity, diabetes, and associated cardiovascular and renal comorbidities are perceived as states of energy overabundance [182]. Consistently, SIRT1 activity is suppressed in these conditions [15,183,184]. In turn, SIRT1 downregulation shifts the HIF-1α/HIF-2α balance in favor of HIF-1α. Conversely, a low energy status activates SIRT1 and promotes HIF-2α over HIF-1α [24,183]. Among the agents with antihyperglycemic properties and/or with cardiorenal protective effects, resveratrol and SGLT2 inhibitors have also been shown to promote SIRT1 activity.

#### 5.2.1. Resveratrol

Resveratrol is a phytochemical with protective properties against diabetes-induced cardiovascular and renal disease (recently reviewed in [185]). The beneficial effect of this natural compound is mainly mediated by indirect antioxidant mechanisms via nuclear factor erythroid 2-related factor 2 activation [186]. Nonetheless, the effect of resveratrol on cellular energetics and metabolism may also play a role. In fact, resveratrol inhibits HIF-1α in diverse cells and tissues [43,187,188] by virtue of its capacity of modulating the activity of SIRT1 [187,188]. Consistently, its myocardial and renal protective effect in acute hypoxic injury and aging has been associated with the suppression of HIF-1α via SIRT1 and antioxidant-related mechanisms [188,189,190]. The effect of resveratrol on HIF-2α has not been studied yet. However, its ability to reduce EPO suppression in diabetes [191] is likely related to an action on HIF-2α via SIRT1 upregulation [192].

#### 5.2.2. Sodium Glucose Co-Transporter 2 (SGLT2) Inhibitors

Other antidiabetic drugs may possibly exert their protective effects against vascular complications by modulating the SIRT1/HIF signaling axis. There is experimental and clinical evidence that SGLT2 inhibitors have cardioprotective and renoprotective effects additional to those resulting from improved glycemic control [184]. Among the several mechanisms proposed to explain these protective extraglycemic effects (reviewed in [193]), a very interesting one is that SGLT2 inhibitors may act by regulating the activity of the energy/redox sensing molecule SIRT1 (Figure 6).

By reflecting the enhanced delivery of glucose to the kidney, the increased expression and activity of SGLT2 may be considered itself a biomarker of nutrient overabundance [194]. Anyway, SGLT2 overexpression is accompanied by SIRT1 downregulation in kidneys of diabetic mice and humans [195]. What is more, SGLT2 upregulation may serve as a “fuel-burning system” that promotes a metabolic shift to aerobic glycolysis (i.e., the Warburg effect) in proximal tubular cells [196]. SGLT2 inhibitors may counteract these energetic and metabolic changes by inducing a loss of calories in the urine that mimics the effect of nutrient deprivation and triggers a fasting-like transcriptional paradigm in cardiorenal tissues [184]. This comprises the activation of SIRT1 and its downstream effects [194], including HIF-1α downregulation. In support of this assumption, there is preliminary but compelling experimental evidence that SGLT2 inhibitors provide protection against diabetic complications by therapeutically targeting HIF-1α accumulation [160,197,198,199,200]. The normalization of HIF-1α signaling may also explain the effect of SGLT2 inhibitors in restoring energetic and redox alterations induced by diabetes in vascular tissues [197,198,200], including modifications in mitochondrial respiration and glycolysis branch pathways activity [43].

Consistent with the ability to suppress SGLT2 activity and upregulate the SIRT1 pathway [201], SGLT2 inhibitors may protect against cardiovascular and renal complications not only by restraining the HIF-1α response, but also by promoting HIF-2α stabilization and its transcriptional program [24,181,183]. According to this hypothesis, SGLT2 inhibitor therapy increases EPO production in patients with type 2 diabetes and stage 2 or 3 CKD [202]. However, contrary to this hypothesis, the only study that investigated the effect of SGLT2 inhibition on HIF-1α and HIF-2α showed that the amelioration of DKD is associated with the suppression of the abnormal expression of both HIF-α isoforms in glomeruli and tubules in the absence of signs of oxidative stress and/or hypoxia [161]. Unfortunately, this study did not investigate the specific roles of HIF-1α and HIF-2α in the onset and progression of DKD. However, chronological consistency between the overexpression/suppression of HIFs and development/amelioration of glomerular and tubular damage suggests that the prolonged expression of both HIF-α isoforms could be pathologic and that their suppression may be involved in the protection provided by SGLT2 inhibition.

In summary, despite the large number of experimental studies on the beneficial effects of resveratrol and SGLT2 inhibitors in diabetic vascular complications, only few have investigated their effects on HIF-1α, of which only one has also evaluated HIF-2α. Considering that resveratrol and SGLT2 inhibitors should favor the activity of HIF-2α over that of HIF-1α by inducing SIRT1 activity, and that in cardiovascular and renal complications of diabetes the activity of SIRT1 is suppressed [24,195], this is another research area that deserves further study.

## 6. Conclusions

Redox and HIF signaling pathways interact with each other in many ways and at different levels during the course of diabetes and its cardiovascular and renal complications. PHD proteins, the main modulators of HIF stability, directly sense oxygen concentrations, but HIF signaling can also be regulated in a redox-sensitive manner under normoxic conditions. Although the variation of ROS levels has an important impact on HIF proteins stabilization in physiological and pathological conditions, the sources of ROS in diabetes and the kinetics and timing of their production is still a topic of hot debate. Therefore, before hypothesizing an involvement of HIFs secondary to oxidative stress in the pathogenesis of diabetic tissue damage, it is first necessary to understand whether oxidative stress is really the initial instigator of vascular complications or a downstream consequence of earlier molecular modifications (recently reviewed in [5]).

Although the short-term actions of HIF-α proteins can protect against hypoxia-related injury, prolonged HIF signaling (particularly through HIF-1α), leads to increases in inflammation, oxidative stress, and fibrosis [24,151,203]. Several metabolic and local tissue factors typical of diabetes and related metabolic disorders have been shown to affect HIF signaling, also independently of increased ROS production. Changes in the levels of any of these metabolic factors can directly and chronically activate the HIF response and promote changes in cellular metabolism and energetics (i.e., the Warburg effect) [125], leading to the activation of ROS-producing enzymes and proinflammatory/profibrotic pathways [5,43].

HIFs are involved in the maintenance of redox homeostasis in response to acute changes in oxygen or energy substrate availability. However, during sustained activation, as in the case of chronic exposure to metabolic stressors, HIF-1α and HIF-2α may exert mutually antagonistic effects on the redox state and proinflammatory/profibrotic pathways [135,136]. Currently, some evidence suggests that an imbalance in the HIF-α isoforms (elevated HIF-1α and/or reduced HIF-2α) may contribute to the development and progression of cardiovascular and renal complications of diabetes, although direct proof is lacking. Contrary to this hypothesis, recent experimental findings suggest a detrimental role of both HIF isoforms in DKD [161].

In addition to inducing HIF signaling in the early stages [17,18,43,88,116,117], hyperglycemia has also been reported to destabilize HIF-1α protein, causing an impaired hypoxic response (recently reviewed in [13,14]). In fact, tissues affected by diabetic complications are hypoxic [204,205], but it is unclear whether this condition is an early event driven directly by hyperglycemia or, rather, is secondary to vascular dysfunction induced by upstream molecular, metabolic, and biochemical changes (Figure 4). In the latter case, the defective response to hypoxia could likely play a role in the progression rather than the development of diabetic complications. In support of this assumption, HIF-α activation by PHD inhibition has been shown to have opposite effects in the development of CKD depending on the timing of administration and the isoform of HIF-α that is activated. Even more important, recent experimental evidence indicates that hyperglycemia sensitizes kidneys to hypoxia-induced fibrosis via HIF-1α upregulation [173]. Based on this finding, pharmacological inhibitors of HIF-1α may be more indicated than PHD inhibitors for the treatment of diabetic complications. Altogether, these observations warrant caution and further experimental research on the use of PHD inhibitors in diabetic patients, particularly for the prevention and early intervention strategies for vascular complications. At the present state of the art, SGLT2 inhibitors seem more appropriate for this purpose, as their use for the prevention of cardiorenal complications is supported by robust data from clinical trials and real-world evidence [206]. This class of antidiabetic drugs not only has cardioprotective and renoprotective properties through both glycemic and extraglycemic mechanisms [193], but also seems to have a unique ability to modulate HIF response to favor the protective action of HIF-2α over the proinflammatory and prooxidative effects of HIF-1α [24]. However, further mechanistic and translational research is required for this biologically plausible hypothesis to evolve into a solid body of knowledge.

To conclude, the effect of HIF activation on vascular complications of diabetes and the exact role of oxidative stress in the pathogenesis of hyperglycemic damage are still controversial. In this complicated context, studies aimed at understanding the specific mechanisms leading to dysregulated HIF signaling, redox imbalance, and their mutual interactions in cardiovascular and renal complications of diabetes represent a promising field of research and an opportunity for the development of novel therapies.

## Figures and Tables

**Figure 1 antioxidants-11-02183-f001:**
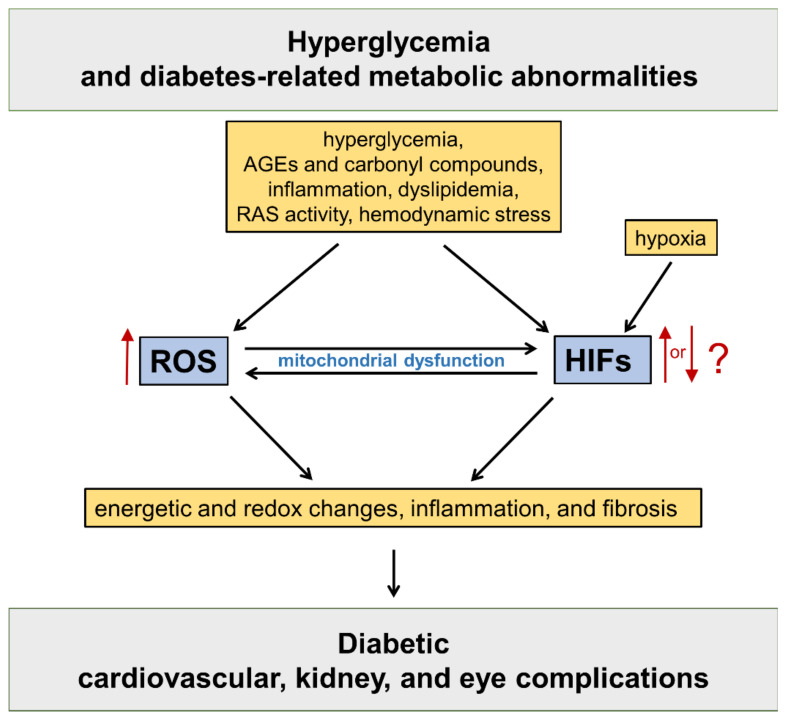
Oxidative stress and hypoxia-inducible factors (HIFs) in the pathogenesis of diabetic vascular complications. Refer to the main text for detailed description. AGEs = advanced glycation end products; RAS = renin angiotensin system; ROS = reactive oxygen species.

**Figure 2 antioxidants-11-02183-f002:**
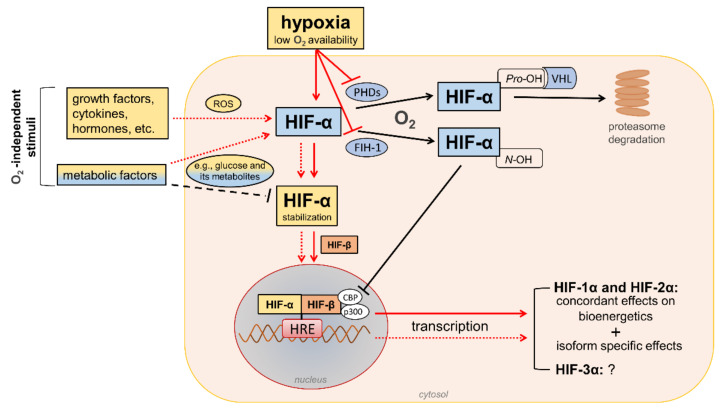
Regulation of hypoxia-inducible factor (HIF) signaling by oxygen (O_2_)-dependent and O_2_-independent mechanisms. There are conflicting opinions on the effects of metabolic factors, particularly hyperglycemia, on the stability of HIF-α proteins (mainly HIF-1α). Refer to the main text for detailed description. Black arrows/lines and light blue boxes/ovals indicate components and processes of the HIF-α degradation pathway in oxygenated cells (black solid arrows/lines) or metabolic factors that may impair HIF-α stabilization in hypoxic conditions (black dashed line). Red arrows/lines and yellow boxes indicate oxygen-dependent (red solid arrows/lines) and oxygen-independent stimuli (red dotted lines) promoting the stabilization and transcriptional activity of HIF-α proteins. ROS = reactive oxygen species; PHDs = prolyl hydroxylase domain proteins; FIH-1 = factor-inhibiting HIF-1; Pro-OH = proline hydroxylation; N-OH = asparagine hydroxylation; VHL = von Hippel–Lindau; CBP = cAMP response element-binding protein; p300 = E1A binding protein p300; HRE = hypoxia-response element.

**Figure 3 antioxidants-11-02183-f003:**
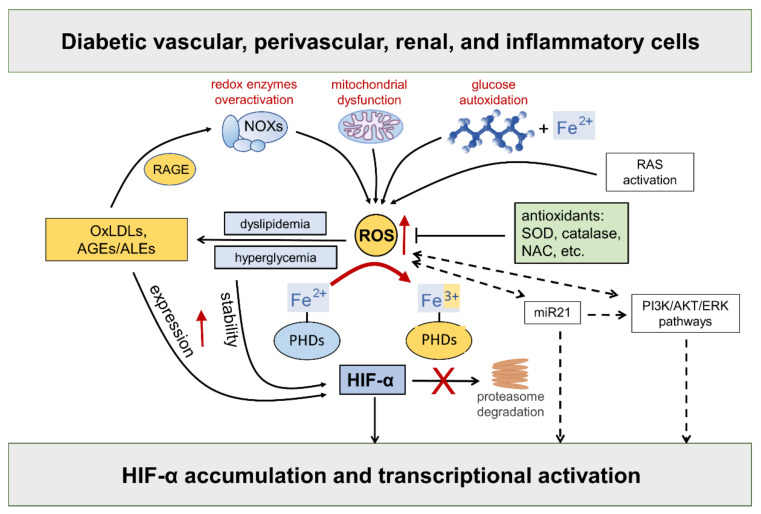
Hypoxia-inducible factor (HIF)-α stability is sensitive to redox status in normoxia: role of diabetes-related stimuli. Refer to the main text for detailed description and references. Dashed arrows = regulation of HIF-α by ROS via indirect mechanisms. NOXs = NADPH oxidases; RAGE = receptor for AGEs; RAS = renin angiotensin system; AGEs = advanced glycation end products; ALEs = advanced lipoxidation end products; OxLDLs = oxidized Low-Density Lipoprotein; ROS = reactive oxygen species; SOD = superoxide dismutase; NAC = N-acetyl-L-cysteine; PHDs = prolyl hydroxylase domain enzymes; miR21 = microRNA 21; PI3K = phosphoinositide 3-kinase; Akt = Ak strain transforming; ERK = extracellular signal-regulated kinase; PHDs = prolyl hydroxylase domain enzymes.

**Figure 4 antioxidants-11-02183-f004:**
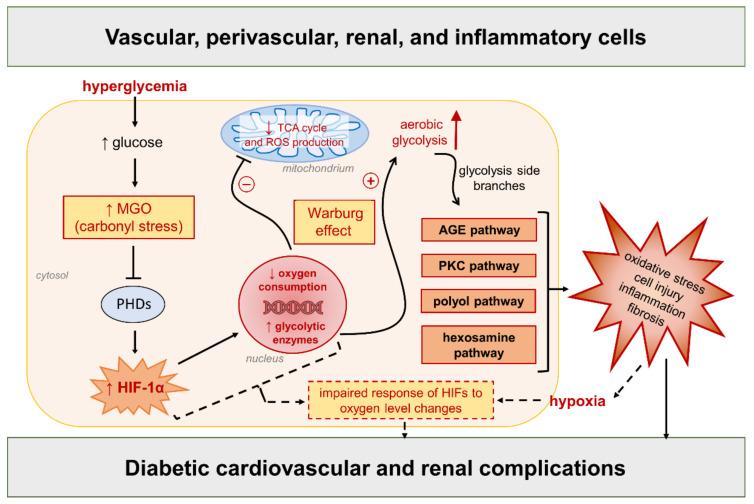
Link between glucose-derived carbonyl stress and the Warburg effect as a new unifying hypothesis of diabetic complications [5,43], alternative to the overproduction of mitochondrial reactive oxygen species (ROS) induced by hyperglycemia. Refer to the main text for detailed description and references. Dashed arrows/lines/box = subsequent processes (or “second hit”) participating in the progression of diabetic complications. TCA = tricarboxylic acid; MGO = methylglyoxal; AGE = advanced glycation end product; PKC = protein kinase C; PHDs = prolyl hydroxylase domain enzymes; HIF-1α = hypoxia-inducible factor-1α.; HIFs = hypoxia-inducible factors.

**Figure 5 antioxidants-11-02183-f005:**
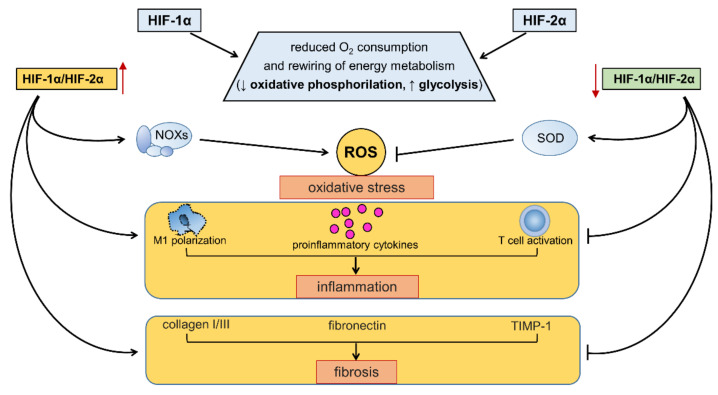
Effects of hypoxia-inducible factor (HIF)-1α, HIF-2α, and changes in HIF-1α/HIF-2α balance on energy metabolism, redox balance, inflammation, and fibrosis. Refer to the main text for detailed description and references. NOXs = NADPH oxidases; ROS = reactive oxygen species; SOD = superoxide dismutase; M1 = classically activated macrophages.

**Figure 6 antioxidants-11-02183-f006:**
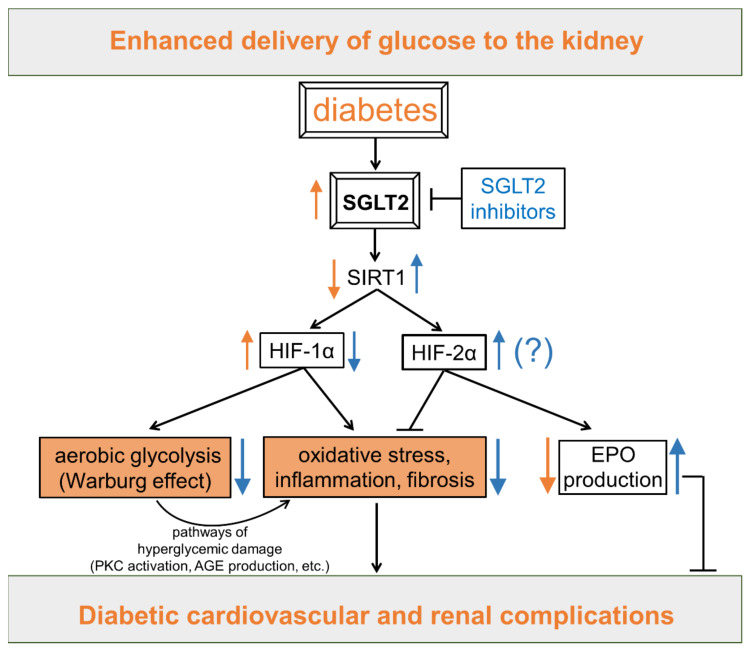
Potential cardioprotective and renoprotective effects of sodium glucose co-transporter 2 (SGLT2) inhibitors through modulation of sirtuin-1 (SIRT1) and the hypoxia-inducible factor (HIF) system. Refer to the main text for detailed description and references. Orange arrows/boxes are effects induced by diabetes. Blue arrows are effects induced by SGLT2 inhibitors. (?) = grounded on a solid rationale, but yet to be demonstrated; PKC = protein kinase C; AGE = advanced glycation end product; EPO = erythropoietin.

## Data Availability

Not applicable.

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
