# Peer review of "Mutual Regulation between Redox and Hypoxia-Inducible Factors in Cardiovascular and Renal Complications of Diabetes"

_antioxidants, 2022, doi:10.3390/antiox11112183_

Round 1

Reviewer 1 Report

Detailed, scholarly, extensively referenced review connecting the cellular redox state and ROS to the activation hypoxia cellular signaling.

Author Response

Reviewer #1

Detailed, scholarly, extensively referenced review connecting the cellular redox state and ROS to the activation hypoxia cellular signaling.

We thank the Reviewer for her/his overall positive judgement

Reviewer 2 Report

In the review article by Iacobini and colleagues, the authors summarized the role of oxidative stress and its inducible factors in the kidney-heart axis in diabetes. The contribution of reactive oxygen species (ROS) in the clinical manifestation of multiple diseases is well characterized and studied by focusing on molecular mechanisms including but not limited to inflammation, immunomodulation, dyslipidemia, visual dysfunctions, diabetes, carcinogenesis, cardiovascular diseases, and renal and pulmonary complications. ROS covers a wide range of physiological processes. This well-orchestrated review article is suitable for publication in the correct format. If the authors include the renin-angiotensin system and its contribution to renal and cardiac diseases in diabetes, it will surely increase its impact and readability.

Author Response

Reviewer #2

In the review article by Iacobini and colleagues, the authors summarized the role of oxidative stress and its inducible factors in the kidney-heart axis in diabetes. The contribution of reactive oxygen species (ROS) in the clinical manifestation of multiple diseases is well characterized and studied by focusing on molecular mechanisms including but not limited to inflammation, immunomodulation, dyslipidemia, visual dysfunctions, diabetes, carcinogenesis, cardiovascular diseases, and renal and pulmonary complications. ROS covers a wide range of physiological processes. This well-orchestrated review article is suitable for publication in the correct format. If the authors include the renin-angiotensin system and its contribution to renal and cardiac diseases in diabetes, it will surely increase its impact and readability.

We thank the Reviewer for providing us with positive comments and useful suggestions. As requested, we briefly discussed the contribution of the renin-angiotensin system to cardiovascular and renal complications of diabetes in relation to the subject of the review (i.e., redox and hypoxia inducible factors). Please, see lanes 160-163, 374-375, the two new references (#48 and #157), and modified Figure 3 of the revised version.

Reviewer 3 Report

The authors made an actual and comprehensive review about redox and HIFs relationship in cardiovascular and renal complications of diabetes. The references were adequate and cover the various items included in the revision.

The included figures adequately resume the interactions discussed in each section.

The authors should review the following items:

-          Line 40: the authors refer to radical oxygen species. I think that it should be reactive instead radical because there are reactive oxygen species that are not radicals.

-          Figure 2: this figure perhaps needs some reformulation because glycolysis is a cytosol pathway. The fact that the glycolysis is included in the square with mitochondrial respiration can be misleading

-          References sections: some journals names are written in the simplified form. The authors should harmonize and use always the same format.

Author Response

Reviewer #3

The authors made an actual and comprehensive review about redox and HIFs relationship in cardiovascular and renal complications of diabetes. The references were adequate and cover the various items included in the revision. The included figures adequately resume the interactions discussed in each section.

We thank the Reviewer for her/his overall positive judgement and feedback in improving our manuscript

The authors should review the following items:

-          Line 40: the authors refer to radical oxygen species. I think that it should be reactive instead radical because there are reactive oxygen species that are not radicals.

Thank you for reporting this mistake, which we have now corrected.

-          Figure 2: this figure perhaps needs some reformulation because glycolysis is a cytosol pathway. The fact that the glycolysis is included in the square with mitochondrial respiration can be misleading

We have modified Figure 2 according to the Reviewer’s advice

-          References sections: some journals names are written in the simplified form. The authors should harmonize and use always the same format.

Thank you for noticing this issue, which we have now amended